# GENERALIZED VIDEO MOMENT RETRIEVAL

**You Qin**[12*], **Qilong Wu**[23*]

**Yicong Li**[2†], **Wei Ji**[1†], **Shawn Li**[24], **Pengcheng Cai**[5], **Lina Wei**[5], **Roger Zimmermann**[2]

[1]Nanjing University, [2]National University of Singapore, [3]Shanghai AI Laboratory

[4]University of Southern California, [5]Zhejiang University

{qinyou,qilong_wu}@u.nus.edu, lyc071719@gmail.com

weiji@nju.edu.cn, li.li02@usc.edu, rogerz@comp.nus.edu.sg

Code: https://github.com/42xingxing/NExT-VMR

## ABSTRACT

In this paper, we introduce the Generalized Video Moment Retrieval (GVMR) framework, which extends traditional Video Moment Retrieval (VMR) to handle a wider range of query types. Unlike conventional VMR systems, which are often limited to simple, single-target queries, GVMR accommodates both non-target and multi-target queries. To support this expanded task, we present the NExT-VMR dataset, derived from the YFCC100M collection, featuring diverse query scenarios to enable more robust model evaluation. Additionally, we propose BCANet , a transformer-based model incorporating the novel Boundary-aware Cross Attention (BCA) module. The BCA module enhances boundary detection and uses cross-attention to achieve a comprehensive understanding of video content in relation to queries. BCANet accurately predicts temporal video segments based on natural language descriptions, outperforming traditional models in both accuracy and adaptability. Our results demonstrate the potential of the GVMR framework, the NExT-VMR dataset, and BCANet to advance VMR systems, setting a new standard for future multimedia information retrieval research.

## 1 INTRODUCTION

Video moment retrieval (VMR) has rapidly evolved as a pivotal task in the domain of video understanding, garnering significant attention in the fields of information retrieval Ji et al. (2025) and multi-modal understanding Li et al. (2024a; 2023a). At its core, VMR involves identifying a specific segment or moment within a video that corresponds to a given query, usually in the form of a natural language (Zhao et al., 2023; Li et al., 2023b; Ji et al., 2023b;a). VMR is beneficial to a series of downstream tasks, such as video question answering (Li et al., 2022a;b; Xiao et al., 2022; Ji et al., 2025; Li et al., 2023c;d; Xiao et al., 2025), video relationship detection (Shang et al., 2017; 2021; Li et al., 2021), and video dialog (Chu et al., 2021; Pham et al., 2022; Li et al., 2024b), and other multimodal tasks (Liu et al., 2023b; 2025; 2024). This task holds immense potential across various applications, including interactive media, surveillance systems, educational tools, etc. The promise of VMR lies in its ability to refine and expedite the retrieval of video moments based on complex queries, thereby elevating user engagement and paving the way for groundbreaking advancements in multimedia information retrieval technologies.

Despite substantial progress, current VMR models face notable challenges, particularly in handling diverse and complex query scenarios. Traditional VMR approaches typically focus on simple, direct queries that correspond to a single target moment within a video. This restriction limits the practicality of current VMR models, as real-world queries often encompass more nuanced and varied requirements, such as identifying multiple moments from a single query or processing queries with no relevant targets in the video. As shown in Fig. 1, the retrieved video clips generated by VMR models can be multiple (*"There are two dogs positioned in front on a grassy field with a woman nearby"*) or none (*"Two cattle are positioned together, a dog standing on a grassy field nearby"*) in real applications, which means the query can indicate any number of target video clips. Moreover,

---

*Equal contribution. †Corresponding authors.

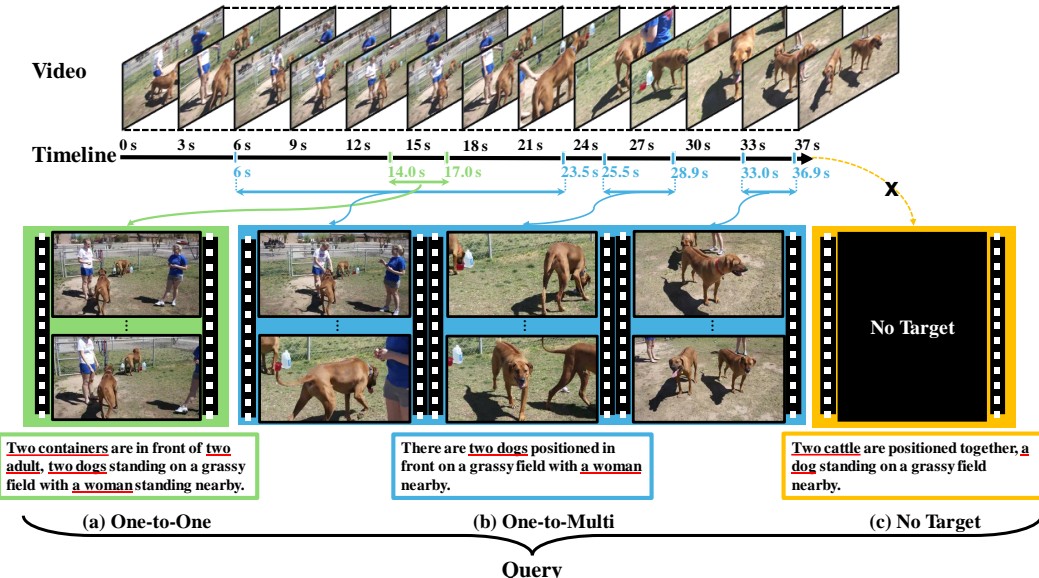

Figure 1: Illustration of Generalized Video Moment Retrieval (GVMR). Each block in different colors represents different video clips and words with red underline represent the salient objects in the clip. Compared with current VMR datasets (a) that only define one clip as ground truth for each query, our GVMR can output description-relevant video clips in various numbers, including multiple (b) or even none (c), which is more in line with real applications.

addressing these challenges necessitates a more generalized VMR model that can handle a broader spectrum of query types, including both one-to-multi and no-target queries, thereby enhancing the model's versatility and applicability in real-world scenarios.

Motivated by these challenges and inspired by Liu et al. (2023a), we introduce an innovative framework termed Generalized Video Moment Retrieval (GVMR). GVMR is designed to adeptly navigate the complexities associated with a multitude of query types, extending the conventional VMR paradigm to encompass both multiple and no-target queries. By harnessing a comprehensive dataset that encapsulates a broad range of query scenarios, GVMR aspires to deliver a robust and flexible solution for video moment retrieval, significantly enhancing the model's interpretative prowess, precision, and retrieval efficiency.

Motivated by this, we build a specialized dataset, NExT-VMR, which is derived from the YFCC100M dataset (Thomee et al., 2016) after meticulous construction and analysis. This dataset is tailored specifically for GVMR, featuring a diverse array of query types, including one-to-multi and no-target queries. The dataset is meticulously labeled and processed to reflect real-world complexities, with a balanced distribution of various query types. This rich dataset serves as the foundation for training and evaluating the VMR model in a generalized scenario, providing a comprehensive understanding of its performance in a more general setting.

Besides, we also propose a new Video Moment Retrieval (VMR) model named BCANet, designed to enhance the performance on the challenging task of VMR, where the model needs to accurately predict variable numbers of temporal timestamps based on natural language descriptions. BCANet leverages a sophisticated transformer encoder-decoder architecture, which harmonizes vision and language features through advanced pre-trained models, thereby creating a unified dimensional space for feature integration. The cornerstone of BCANet is the Boundary-aware Cross Attention (BCA) module, an inventive component that utilizes contrastive learning to produce boundary-aware video features. This module further employs query-region cross attention, ensuring a global and uniform perception of video features by decoder queries, which significantly enhances the precision and accuracy in predicting the start and end times of video segments, along with generating evidence scores for each prediction. The BCA module is particularly notable for its dual components: the Boundary-aware Contrastive Learning (BCL) and the Query-region Cross Attention (QCA). BCL focuses on utilizing ground-truth labels to contrast region features with sentence features, fostering the

generation of video features that are acutely aware of boundary information. This is complemented by QCA, which leverages these enhanced features to ensure that decoder queries have a comprehensive and uniform understanding of the video content, enabling precise retrieval of relevant video moments.

In summary, our contributions are summarized below:

- We first introduce a novel Generalized VMR task that significantly expands the scope of traditional VMR, catering to a wider range of query types and enhancing the model's practical utility.

- We present a meticulously curated dataset, NExT-VMR, specifically designed for GVMR task, which includes a variety of query types and scenarios, thereby enabling more robust and versatile model training and evaluation.

- To deal with GVMR, we propose the BCANet model, which incorporates the innovative Boundary-aware Cross Attention and Query-region Cross Attention modules. This model is underpinned by the Boundary-aware Contrastive Learning strategy, enabling effective modeling of interactions and flexible adaptation to diverse video features.

## 2 RELATED WORKS

### 2.1 VIDEO MOMENT RETRIEVAL DATASETS

Current video moment retrieval (VMR) datasets vary in their focus, ranging from everyday activities to specific scenarios, each contributing uniquely to the advancement of VMR models. Here we summarize the key features and challenges presented by notable VMR datasets.

**Diverse Scenarios and Complexities in VMR Datasets:** Datasets like DiDeMo (Anne Hendricks et al., 2017) and TEMPO (Anne Hendricks et al., 2018), originating from vast video collections like YFCC100M , present scenarios of varied human activities. DiDeMo (Anne Hendricks et al., 2017), with its structure of five-second video segments, offers a simplified retrieval challenge with fixed-length segments. TEMPO (Anne Hendricks et al., 2018)extends this by incorporating more complex, human-generated language queries, pushing models towards a better understanding of natural language in VMR. In contrast, Charades-STA (Sigurdsson et al., 2016), derived from the Charades dataset, delves into daily indoor activities, enriching VMR with semi-automatically generated annotations that align detailed activity labels with video descriptions.

**Challenges in Scale and Annotation Diversity:** ActivityNet Captions (Caba Heilbron et al., 2015), based on the ActivityNet benchmark, stands out for its scale and application in dense video captioning, effectively doubling as a VMR resource. It offers a broad spectrum of human activities, crucial for training models in diverse, dynamic environments. TACoS (Regneri et al., 2013a), focusing on cooking activities, brings scene-specific challenges, combining fine-grained activity labels with natural language descriptions. This specificity in scene offers a unique testbed for VMR models to understand and process scene-specific activities and their linguistic descriptions.

**Advancing VMR with Long-form Content and Unbiased Annotations:** The MAD dataset (Soldan et al., 2022) marks a significant evolution in VMR, shifting towards mainstream, long-form movies. It challenges conventional VMR approaches with its vast scale and nuanced, descriptive queries. Unlike other datasets, MAD addresses the issue of hidden biases in VMR, providing a more accurate and unbiased annotation framework. This shift towards long-form content opens up new avenues in VMR, particularly in understanding and retrieving moments from extensive, continuous video streams.

**Synthesis of Real-world Application and Research:** Collectively, these datasets offer a comprehensive landscape for VMR research, each introducing distinct challenges – from fixed-length segments in DiDeMo to complex, multi-scene activities in ActivityNet Captions, and the narrative-driven, long-form content of MAD. The diversity in video content, query complexity, and annotation methods across these datasets underscores the multifaceted nature of VMR challenges, driving the development of more sophisticated and versatile VMR models. To make VMR closer to real application, we propose the generalized video moment retrieval task with clips in arbitrary numbers as retrieval results for each query in this paper.

# 3 PROPOSED DATASET

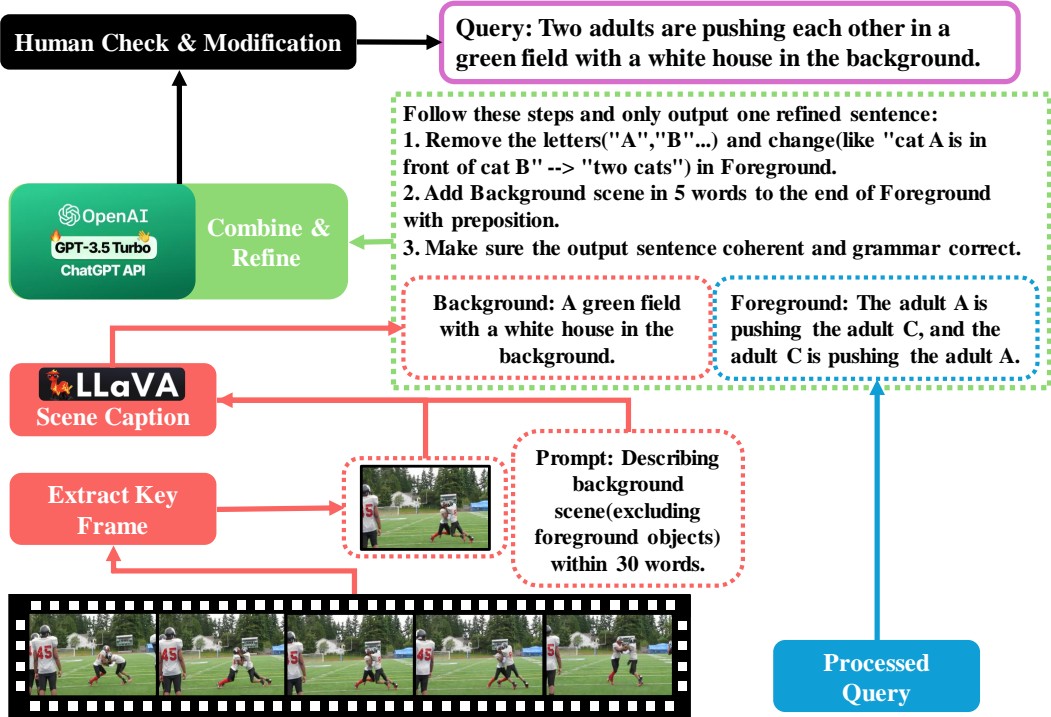

Figure 2: The overall pipeline for Query Generation in NExT-VMR dataset. With the help of current LLM models, such as GPT and LLaVa, we can generate queries in fine-grain semantic, so as to benefit the video content understanding to the next stage.

## 3.1 TASK DEFINITION

This section defines two primary challenges in video moment retrieval tasks: no-target queries and one-to-many queries. These task definitions are crucial for understanding and improving video moment retrieval approaches.

**No-Target.** In no-target queries ($n = 0$), the query statements have no corresponding target moments in the video content. This scenario is common in real-world applications, as user queries may not relate to the video content. The key to evaluating this task is the model's ability to correctly identify these queries as no-target and respond with corresponding empty timestamps.

**One-to-Multi.** In one-to-multi queries, we focus on how a single query statement corresponds to multiple moments in a video. Unlike traditional retrieval tasks which map a query to only one moment, this task reflects a more complex and realistic scenario where a single query may pertain to several relevant segments within the content. This reflects real use cases where an inquiry could span multiple, non-consecutive events or scenes that share a thematic or narrative link. The evaluation of such queries necessitates a model that can not only retrieve multiple distinct moments but also understand the broader context that binds these moments together. Success in this task is indicative of a model's advanced comprehension capabilities, allowing it to provide a comprehensive response that encapsulates all pertinent instances within the video, thereby enhancing the user's search experience by acknowledging the multifaceted nature of their request.

## 3.2 DATA SOURCE

We utilizes the YFCC100M [1] dataset as the primary data source. As the largest publicly collection, the YFCC100M dataset offers a wealth of video content. We extend its application to video moment retrieval tasks, focusing particularly on one-to-many queries and no-target queries.

## 3.3 DATA PREPARATION

We manually annotated the original YFCC100M video collection with $(subject, predicate, object)$ relation tuples and their corresponding timestamps $\{(t_s, t_e)\}$, then transformed the tuples into grammatically correct sentences to serve as query statements. Here, $n$ indicates the number of correct relation tuples in a query statement. Next, we create arbitrary combinations of relation tuples for $n = 2$ and $n = 3$ scenarios. For instance, for $n = 3$, we intersect the timestamps of three relation tuples to form a new query statement. These query statements are used to match corresponding moments in videos.

## 3.4 DATA LABELING & PROCESSING

### 3.4.1 NO-TARGET GENERATION

In constructing the No-Target part of the dataset, we specifically generated such query statements using the following methods:

**Randomly Modifying Query Statements.** We select $(s, p, o)$ relation tuples from $n = 1$ scenarios, randomly modifying the subject ($s$) or object ($o$) to make them unrelated to the video content. We also randomly change the predicate ($p$) to create new query statements.

**Ensuring Grammatical Diversity.** We use a dictionary mapping to ensure the generated query statements are both grammatically correct and incorrect, approximately in a 1:1 ratio, adding complexity to the task.

**Cross-Video Query Statements.** Other query statements come from non-target videos, i.e., $n = 1$ query statements randomly extracted from other videos in the dataset.

### 3.4.2 ONE-TO-MULTI GENERATION

When constructing the One-to-Multi part of the dataset, we followed these steps:

**Constructing Single Relation Pair Queries ($n = 1$).** We first extract $(s, p, o)$ relation tuples from the original YFCC100M dataset and convert them into grammatically correct sentences, forming single relation pair queries. Here, $n = 1$ indicates that a query statement contains only one correct relation tuple, which can correspond to a precise moment in the video (i.e., the ground truth moment's timestamps).

**Combining Query Statements ($n > 1$).** Next, we combine different single relation pair queries to create $n = 2$ and $n = 3$ scenarios. Specifically, we look for intersections in the timestamps of relation tuples from different query statements. For example, in the case of $n = 3$, there are three original relation pairs, and their respective ground truth timestamps are $(t_s^1, t_e^1)$, $(t_s^2, t_e^2)$ and $(t_s^3, t_e^3)$.

When these relation pairs have a temporal intersection, we define this intersection as $(t_s', t_e')$ In this case, we combine these relation pairs into a new query statement and define it as the $n = 3$ scenario.

Moreover, during the pairing phase, we implement a 5% threshold to curtail the creation of intersecting queries originating from an abundance of ephemeral moments. This measure is instituted to diminish the potential for superfluous noise within the query set. In addition, we synthesize the extracted subject-predicate-object $(s, p, o)$ relational pairs by amalgamating multiple intersecting pairs of $(s, p, o)$ using the conjunction ",and". This approach facilitates the construction of more streamlined queries that cohesively represent the combined elements of the foreground.

---

[1]Videos from YFCC100M (Thomee et al., 2016) are directly crawled from Flickr.

### 3.4.3 Scene Caption & Query Refinement with LLM

Figure 2 illustrates a detailed overview of our whole query generation and refinement pipeline. These initial queries are derived from a rich, contextually-aware scene graph generated for each video. To enhance the descriptiveness and semantic richness of the background scene graph, we employ the advanced llava-v1.5-7b model with Int8 Quantization, which is renowned for its efficiency and accuracy in scene understanding and caption generation tasks.

Subsequent to the background scene graph captioning, we integrate the scene's background information with the dynamic foreground elements. This holistic approach ensures that each query encapsulates a comprehensive view of the video content, accounting for both the primary subjects of interest in the foreground and the contextual details provided by the background.

The final and critical phase of our pipeline involves the refinement of these merged queries. For this purpose, we leverage the powerful natural language processing capabilities of the gpt-3.5-turbo API. The API's nuanced understanding of language nuances allows us to polish the queries to meet high standards of clarity and relevance, thus greatly enhancing the overall quality and effectiveness of our video moment retrieval system as depicted in the process flow of Figure 2. Additional statistical analyses of the dataset are provided in Section A.1.

Table 1: Comparison of NExT-GVMR with various existing VMR datasets. Two columns of Multi-target and No-target indicate if the dataset could be used for Multi-target or No-target Moment Retrieval. * indicates that the senario which excludes no target queries.

| Dataset | # Videos | # Queries | Avg. # Query/Video | Avg. # Seg/Query | Domain | Multi-target | No-target |
|---|---|---|---|---|---|---|---|
| ANet-Captions (Caba Heilbron et al., 2015) | 14,926 | 71,953 | 4.83 | 1 | Open | ✗ | ✗ |
| Charades-STA (Sigurdsson et al., 2016) | 9,848 | 27,847 | 2.83 | 1 | Indoor activities | ✗ | ✗ |
| DiDeMo (Anne Hendricks et al., 2017) | 10,464 | 40,543 | 3.87 | 1 | Open | ✗ | ✗ |
| TEMPO (Anne Hendricks et al., 2018) | 10,464 | - | - | 1 | Open | ✗ | ✗ |
| MAD (Soldan et al., 2022) | 650 | ≈ 384,600 | ≈ 592 | 1 | Movies | ✗ | ✗ |
| QVHighlights (Lei et al., 2021) | 10,148 | 18,367 | 1.02 | 1.8 | Vlog / News | ✗ | ✗ |
| TACoS (Regneri et al., 2013a) | 127 | 18,818 | 143.52 | 1 | Cooking | ✗ | ✗ |
| NExT-VMR (Ours)* | 9,957 | 123,238 | 12.38 | 1.88 | Open | ✓ | ✗ |
| NExT-VMR (Ours) | 9,957 | 153,191 | 15.39 | 1.51 | Open | ✓ | ✓ |

### 3.5 Dataset Comparison

Finally, we compared our dataset with several other popular video moment retrieval datasets. This comparison mainly focuses on data diversity, difficulty, and practical application value. As shown in the Table 1, our dataset, GVMR, stands out in several aspects. Unlike other datasets, GVMR has two versions: one with a focus on multi-target instances and another that includes no-target instances, enhancing the complexity and variability in video moment retrieval tasks. GVMR contains 9,957 videos with over 123,238 queries in one version and 153,191 in another, indicating a substantial number of unique query-video pairings. This volume is comparatively higher than most datasets listed, with the exception of MAD, which has a high number of queries per video due to its movie domain focus. However, GVMR provides a more extensive average number of segments per query, which can be more challenging for retrieval algorithms. Moreover, GVMR's domain is open, making it applicable to a wide range of real-world scenarios, unlike others that are confined to specific domains such as indoor activities, cooking, or movies. The design of GVMR, which includes multi-target and no-target instances, not only increases its relevance for practical applications but also sets a higher bar for the development of video moment retrieval systems by requiring them to discern more nuanced aspects of video content.

## 4 Proposed Model

In the GVMR task, the input comprises an untrimmed video, represented as $\mathbf{V} = \{\mathbf{v}_i\}, i \in \{1, \ldots, M\}$, and a natural language description, denoted as $\mathbf{W} = \{\mathbf{w}_j\}, j \in \{1, \ldots, N\}$, where $M$ and $N$ indicate the total number of frames in the video and words in the description, respectively. The objective is to accurately predict a set of temporal timestamps $\{(t_s^k, t_e^k)\}, k \in \mathbb{Z}$, with $t_s$ and

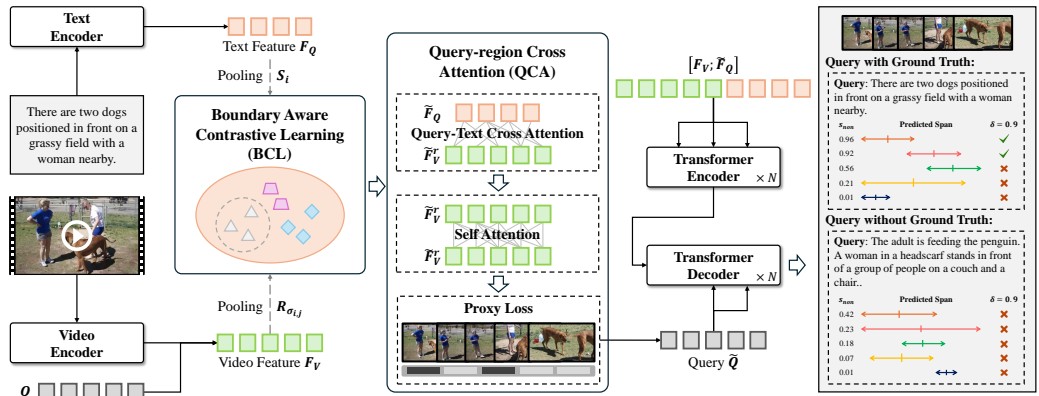

Figure 3: The architecture of our proposed BCANet, primarily comprising the Boundary-aware Contrastive Learning and Query-region Cross Attention modules. These components enable learnable queries to uniformly perceive and analyze intricate details within video segments, facilitating robust time span predictions. Additionally, confidence scores are utilized for ranking final predictions and filtering in no-target detection samples.

$t_e$ representing the start and end timestamps, respectively. Unlike previous approaches, this task introduces an increased level of complexity as the number of predicted timestamps is variable. This means that a natural language description may correspond to none, one, or multiple segments within the video. Our evaluation metrics penalize both over-prediction and under-prediction of target objects, underscoring the need for precise temporal localization in response to the given descriptions.

## 4.1 MODEL OVERVIEW

To tackle the GVMR task more effectively, we employ a transformer encoder-decoder model (Figure 3). The video $V$ is processed by two pre-trained models—SlowFast (Feichtenhofer et al., 2019) and the CLIP video encoder (ViT-B/32)—to extract vision features $F_V \in \mathbb{R}^{m \times C_v}$, where $m$ is the temporal length and $C_v$ the feature dimension. For language input, the CLIP text encoder extracts token features $F_Q \in \mathbb{R}^{n \times C_q}$ (with $n$ tokens and $C_q$ as the hidden dimension). Two linear layers then align these features to a unified dimension $C$, yielding $F_V \in \mathbb{R}^{m \times C}$ and $F_Q \in \mathbb{R}^{n \times C}$. Following (Lei et al., 2021), concatenated video and sentence features are fed into a transformer encoder to produce hidden features $H$ that interact with decoder queries via cross-attention for boundary prediction. These queries are generated by the Boundary-aware Cross Attention (BCA) module (Section 4.2), where $F_Q$ is average-pooled and refined by MLPs to yield the sentence feature $S$; combining $S$ with boundary features from $F_V$ forms positive and negative pairs for boundary-aware contrastive learning. Subsequently, $L$ learnable queries interact with the enhanced features via query-region attention, supervised by a proxy loss to ensure accurate local perception. The final queries predict start and end times $\mathcal{T} = \{(t_s^k, t_e^k)\}$ and evidence scores $e^k$ through a three-layer MLP. No-target detection uses a threshold $\delta$, classifying samples as no-target if $\forall e \in \{e^k\}, e < \delta$. Both $\mathcal{T}$ and evidence scores are supervised by the matching cost introduced in (Carion et al., 2020).

## 4.2 BOUNDARY-AWARE RELATION MODELING

This section introduces the Boundary-aware Cross Attention (BCA) module, which combines two key components: (1) Boundary-aware contrastive learning, leveraging ground-truth labels to produce boundary-aware video features, and (2) Query-region cross attention, ensuring decoder queries uniformly perceive these enhanced features for more accurate predictions.

**Boundary-aware Contrastive Learning (BCL):** Unlike previous boundary modeling approaches (Wang et al., 2019), which directly perform binary boundary classification on the features before grounding MLP, our component leverages ground-truth labels to aggregate region features and

contrast these with sentence features to generate boundary-aware video features. Specifically, it utilizes ground-truth label information $(\tilde{t}_s^k, \tilde{t}_e^k)$ to collate video features within each labeled duration. The feature indices in the temporal space are denoted as $(\tilde{t}_{s/e}^k)$. By applying average pooling, a region feature unit $(R_{\sigma_{i,j}})$ is obtained, formulated as:

$$R_{\sigma_{i,j}} = \text{AvgPooling}([F_V^{\tilde{t}_s^k}, \cdots, F_V^{\tilde{t}_e^k}]) \tag{1}$$

Here, $[\cdot]$ represents the concatenation operation, and $\sigma$ indicates the index corresponding to the $j$-th ground-truth of the $i$-th sample in a training batch. For non-target samples, the entire video feature set is utilized as a negative sample:

$$R_{\sigma_{i,0}} = \text{AvgPooling}(F_V) \tag{2}$$

Subsequently, the sentence feature $S_i$ is contrasted with the above-mentioned region features to stretch the feature space for contrastive learning. In each batch, the positive sample is the region features $\{R_{\sigma_{i,j}}\}$ corresponding to the sentence $S_i$'s ground-truth label, with other batch indices forming negative pairs. This forms the basis for symmetric contrastive learning loss calculation:

$$\mathcal{L}_c^{R \to S} = -\log \frac{\sum_j \exp\left(\cos\left(R_{\sigma_{i,j}}, S_i\right)/\tau\right)}{\sum_{i=1}^n \sum_j \exp\left(\cos\left(R_{\sigma_{i,j}}, S_i\right)/\tau\right)} \tag{3}$$

$$\mathcal{L}_c^{S \to R} = -\log \frac{\sum_j \exp\left(\cos\left(S_i, R_{\sigma_{i,j}}\right)/\tau\right)}{\sum_{i=1}^n \sum_j \exp\left(\cos\left(S_i, R_{\sigma_{i,j}}\right)/\tau\right)} \tag{4}$$

Here, $\tau$ denotes the exponential temperature that controls the sharpness of the loss function. Notably, for no-target sample, positive sample contrastive learning loss is not computed due to the absence of positive samples.

**Query-region Cross Attention (QCA):** After enhancing features through contrastive learning to obtain $\tilde{F}_V$, after the further cross-attention with text word-level feature $\tilde{F}_Q$ and self-attention to produce $\tilde{F}_V^r$ the QCA mechanism applies cross-attention between $\tilde{F}_V$ and $L$ learnable queries $Q \in \mathbb{R}^{L \times C}$. Attention maps are computed as:

$$A_{li} = \text{softmax}(Q\sigma(\tilde{F}_V W_{ij})^T) \in R^{L \times m} \tag{5}$$

where $\sigma$ is the GELU activation function. These $L$ span attention maps focus on different video regions, guided by a proxy loss. Region-aware features, $\tilde{Q}_l = A_{li}\sigma(\tilde{F}_V W_{ij})$, which, alongside $F_V$ and $\tilde{F}_Q$, is input to the transformer decoder. The proxy loss ($\mathcal{L}_{proxy}$) identifies queries as positive ($y_l = 1$) or negative based on alignment with ground-truth spans:

$$\mathcal{L}_{proxy} = \frac{1}{L} \sum_{l=1}^L \sum_{i=1}^n -\log(y_i p(A_{li})) \tag{6}$$

This guides the decoding queries to learn features from correct spans, improving prediction accuracy.

## 4.3 LOSS FUNCTION

To facilitate moment localization, we employ a set prediction loss, grounded in bipartite matching as outlined in (Lei et al., 2021). This process involves calculating the optimal assignment based on the similarity of timestamps and their respective confidence scores using the Hungarian algorithm. The ground truth moment timestamps, denoted as $\hat{\mathcal{T}_i} \in [0, 1]^2$, consist of normalized center coordinates and width. The optimal assignment is determined as follows:

Table 2: Comparison of results of various SOTA models and BCANet on the NExT-VMR dataset, where $\mu$ represents the IoU thresholds to categorize a predicted moment as "correct". The **bold** value denotes the highest performance in each prediction type category for each metric. "*" indicates that non-target scenarios are ignored, considering only queries with at least one ground-truth annotation.

| Model | Pred. Type | R1@$\mu$ | | | mIoU | mAP@$\mu$ | | | Avg. | N-acc. | T-acc. |
|---|---|---|---|---|---|---|---|---|---|---|---|
| | | $\mu=0.3$ | $\mu=0.5$ | $\mu=0.7$ | | $\mu=0.25$ | $\mu=0.5$ | $\mu=0.75$ | | | |
| Moment-DETR (Lei et al., 2021) | Multi | 56.75 | 41.86 | 30.36 | 44.41 | 62.90 | 47.03 | 32.43 | 47.45 | 75.12 | 94.28 |
| QD-DETR (Moon et al., 2023) | Multi | 62.27 | 47.70 | 35.10 | 48.94 | 69.98 | 53.87 | 37.06 | 53.64 | 77.68 | 92.67 |
| EaTR (Jang et al., 2023) | Multi | 63.85 | 44.81 | 34.50 | 49.44 | **72.94** | 55.17 | 36.91 | 55.67 | 76.44 | 95.68 |
| Chat-UniVi* (Jin et al., 2024) | Prompt | 57.16 | 35.49 | 17.81 | 38.64 | 62.88 | 35.49 | 14.14 | 37.50 | - | - |
| Chat-UniVi (Jin et al., 2024) | Prompt | 46.15 | 27.52 | 13.81 | 29.62 | 42.13 | 23.71 | 9.95 | 25.26 | 71.55 | 68.48 |
| **BCANet (Ours)** | Multi | **65.06** | **48.61** | **37.10** | **50.24** | 72.75 | **56.29** | **41.22** | **56.75** | **77.92** | **95.79** |

$$\hat{\sigma}' = \arg\min_{\sigma' \in \mathfrak{G}_N} \sum_i^N [-\lambda_c p_{c,\sigma'(i)} + \mathcal{C}(\hat{\mathcal{T}}_i, \mathcal{T}_{\sigma'(i)})], \tag{7}$$

$$\mathcal{C}(\hat{\mathcal{T}}_i, \mathcal{T}_{\sigma'(i)}) = \lambda_{l_1} t \|\hat{\mathcal{T}}_i - \mathcal{T}_{\sigma'(i)}\|_F^1 + \lambda_{iou} \mathcal{L}_{iou}(\hat{\mathcal{T}}_i, \mathcal{T}_{\sigma'(i)}), \tag{8}$$

where $\lambda_{l_1}, \lambda_{iou}$, and $\lambda_c$ are balancing parameters. Using the optimal assignment $\hat{\sigma}'$, the moment localization loss is defined as:

$$\mathcal{L}_{moment} = \sum_i^N [-\lambda_c \log p_{c,\hat{\sigma}'(i)} + \mathcal{C}(\hat{\mathcal{T}}_i, \mathcal{T}_{\hat{\sigma}'(i)})] \tag{9}$$

The overall objective is then formulated as:

$$\mathcal{L}_{overall} = \mathcal{L}_{moment} + \lambda_{bcl}\mathcal{L}_{bcl} + \lambda_{proxy}\mathcal{L}_{proxy}, \tag{10}$$

In this formula, $\lambda_{bcl}$ and $\lambda_{proxy}$ are additional balancing parameters, and $\mathcal{L}_{bcl} = \mathcal{L}_c^{R \to S} + \mathcal{L}_c^{S \to R}$. This comprehensive approach ensures a balanced and effective mechanism for moment localization within the context of the GVMR task.

**Inference.** During the inference phase, the computation of both Boundary-aware Contrastive Learning (BCL) and Proxy predictions will not be conducted.

## 5 EXPERIMENT

### 5.1 EVALUATION METRICS

We mainly evaluate the performance with the following metrics:

**R1@IoU and mAP@X.** We use R1@IoU and mAP@X as primary metrics. For multi-moment retrieval, mAP is averaged over IoU thresholds [0.25, 0.5, 0.75] (following (Caba Heilbron et al., 2015)). For single moment retrieval, Recall@1 IoU (R1@$\mu$) counts a prediction as positive if its IoU is at least $\mu$ (with $\mu$ set to 0.3, 0.5, or 0.7).

**N-Accuracy and T-Accuracy.** N-Accuracy (N-acc.) measures no-target identification, where a no-target sample with no prediction is a true positive (TP) and any prediction is a false negative (FN), computed as $N\text{-acc.} = \frac{TP}{TP+FN}$. T-Accuracy (T-acc.) assesses target classification as $T\text{-acc.} = \frac{TN}{TN+FP}$, with TN and FP denoting true negatives and false positives, respectively.

### 5.2 IMPLEMENTATION DETAILS

We utilize the pre-trained SlowFast (Feichtenhofer et al., 2019) and CLIP (Radford et al., 2021) video encoder model to extract video features with sampling rate 0.2, and we extract token-level

text features by a pre-trained CLIP (Radford et al., 2021) text encoder. We uniformly sampled 128 clips from each video as feature input. During training, AdamW (Ilya Loshchilov, 2019) optimizer with weight decay 1e-4 is adopted; the batch size is set at 32 for training and 128 for testing; the hidden dimension $C = 256$. We configured our transformer encoder and decoder with two layers each, denoted as T = 2. The hyperparameter settings were determined as follows: $L = 10, \lambda_c = 4, \lambda_{l_1} = 10, \lambda_{iou} = 1, \lambda_{bcl} = \lambda_{proxy} = 0.1$, for optimal performance. For the no-target threshold we set it as $\delta = 0.7$ which is experimentally balance for target and no-target generalization.

## 5.3 COMPARISON WITH STATE-OF-THE-ART METHODS

Table 2 presents the performance of SOTA methods on the QV-highlight benchmark, re-implemented with a unified feature backbone and default hyperparameters. For Moment-DETR (Lei et al., 2021), QD-DETR (Moon et al., 2023), and EaTR (Jang et al., 2023), we excluded the highlight detection loss and applied the same threshold $\delta$ to filter no-target predictions. These models exhibit strong performance, effectively handling target and no-target samples, but our model's explicit multi-region detection significantly outperforms these baselines, achieving over $5\%$ improvement in overall performance. Additionally, our model marginally surpasses SOTA methods in no-target scenarios. Furthermore, we conducted Video-Large Language Models (Vid-LLMs) zero-shot evaluation (Jin et al., 2024; Maaz et al., 2024; Lin et al., 2024), among which only Jin et al. (2024) supports long video inputs. While Vid-LLMs show potential in single-segment and no-target detection, they struggle with multi-segment predictions, highlighting a critical area for improvement. Detailed results and analysis are included in Appendix A.3.

## 5.4 ABLATION STUDY

**Effectiveness of proposed BCA module.** As illustrated in Table 3, comprehensive ablation experiments were conducted to explore the impact on model performance when either the Boundary-aware Contrastive Learning (BCL) or the Query-region Cross Attention (QCA) module is omitted. Notably, omitting QCA only requires the removal of the proxy loss, as the BCL module is built on QCA. The results in the table clearly show that both modules contribute incrementally to the model's performance. The most substantial improvement is seen when both modules are used together, highlighting their synergistic effect. Further ablation studies on query length and threshold selection can be found in Appendix A.2.

Table 3: Performance Comparison of BCANet With and Without the Boundary-aware Contrastive Learning (BCL) and Query-region Cross Attention (QCA) Modules.

| Model | R1@0.5 | R1@0.7 | mAP@0.5 | mAP@0.7 |
|---|---|---|---|---|
| w/o Both | 42.31 | 30.41 | 47.06 | 33.42 |
| w/o QCA | 46.76 | 36.88 | 55.03 | 38.97 |
| w/o BCL | 46.20 | 35.33 | 54.54 | 36.42 |
| BCANet | **48.61** | **37.10** | **56.29** | **41.22** |

## 6 CONCLUSION

In this paper, we have introduced and developed the concept of Generalized Video Moment Retrieval (GVMR), which aims to handle complex query scenarios such as one-to-many and no-target queries. By constructing and analyzing a comprehensive dataset NExT-VMR, we provided a solid foundation for training and evaluating our model's performance across a variety of real-world query types. Our BCANet model, equipped with advanced techniques like Boundary Aware Cross Attention, demonstrated a significant leap in performance, handling multiple moments and no-target queries with greater accuracy and efficiency. The model's adaptability and robustness were rigorously tested and validated through extensive evaluations, setting a new benchmark in the field of video moment retrieval.

## 7 ACKNOWLEDGEMENT

This work is supported by the Advanced Research and Technology Innovation Centre (ARTIC), the National University of Singapore under Grant (project number: ELDT-RP2).

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

# A APPENDIX

## A.1 STATISTICAL ANALYSIS

We conducted a detailed statistical analysis of the dataset, including the types of query statements, the distribution of moments, and the proportion of query statements with different $n$ values.

**Query Distribution.** In our expanded YFCC100M dataset, queries are designed to cover a range from no-target to multi-target scenarios.

These queries are categorized into four types: no target ($n = 0$), single relation ($n = 1$), double relation ($n = 2$), and triple relation ($n = 3$). Figure 4 shows the distribution of these queries in the train, validation and test sets. This distribution reflects the diversity and balance of the dataset. Notably, one-to-many queries ($n = 2$ and $n = 3$) occupy a significant proportion in the training set, providing rich data for model training to understand and handle more complex query situations.

**Analysis of No Target Queries.** For no-target queries ($n = 0$), we further categorize them into grammatically correct (right) and incorrect (wrong) queries to reflect the diversity of queries in the real world. Figure 4 shows the distribution of these two types of queries in the train, validation and sets. This categorization is particularly important for training and evaluating video moment retrieval models. The model needs to accurately process and differentiate these two types of no-target queries, ensuring a good response to various query types that may be encountered in real-world applications.

**Timestamp Counts of Queries.** As depicted in Figure 5 (a), the data showcases a pronounced long tail distribution in the counts of queries as they relate to the number of ground-truth timestamps. There is a clear trend where queries associated with a smaller number of timestamps dominate the dataset. This trend sharply declines as the number of timestamps increases, indicating that instances with a higher count of ground-truth moments are significantly rarer.

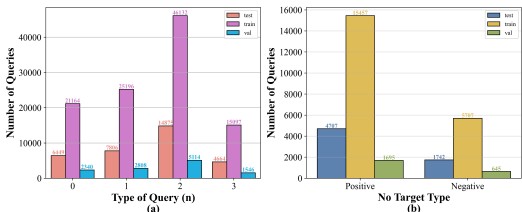

Figure 4: Distribution of (a) query numbers relevant to $n$ clips and (b) Query Type.

The graph on a logarithmic scale highlights the steep drop-off in frequency as the number of timestamps grows, reinforcing the rarity of queries with extensive temporal references. This distribution pattern underlines the challenge in handling queries with multiple timestamps, as they are not only less common but may also represent more complex retrieval scenarios.

**Query Length and Normalized Moment Distribution.** We analyze different percentage of each query length and compare them with Charades-STA, ActicityNet-Captions and TACoS-2DTAN. As shown in Figure 5 (b), the GVMR dataset predominantly consists of queries with a shorter length, with a vast majority falling within the 0-10 words range. This is in contrast to the other datasets, which display a more varied distribution across different query lengths. Charades-STA and ActivityNet-Captions have a more substantial representation in the 10-20 words range, while TACoS-2DTAN features longer queries, with no presence in the shortest category. The concentration of shorter queries in GVMR suggests that the dataset is designed for concise and focused retrieval tasks, which may present a unique challenge for video

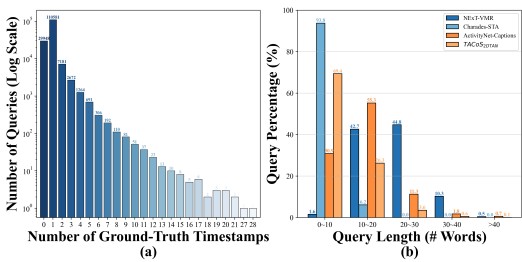

Figure 5: (a) Statistics Distribution for Query Number and Each Corresponding Segments of Ground Truth Timestamps in Log Scale. (b) Distribution for Query Length with comparison with other datasets.

moment retrieval systems that are tuned to datasets with more extended descriptions. The succinct nature of GVMR's queries may require algorithms to be highly efficient at understanding and matching brief textual prompts to video content.

And we also count non-empty normalized ground truth timestamps for each query and visualize the start and end timestamps of each moment. As shown in Figure 6, train and test sets all exhibit similar distributions and share a common characteristic: these moments spread widely across the upper triangle, even though many moments are concentrated in the upper-left region, *i.e.*, a long tail distribution exists among them.

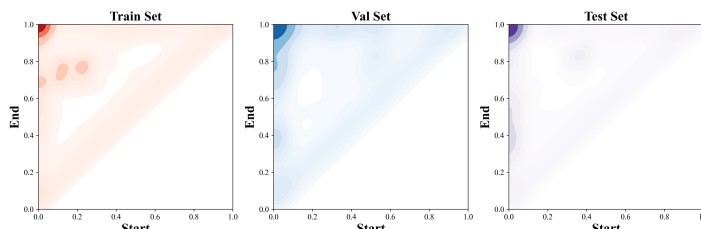

Figure 6: Moment annotation distributions of NExT-VMR dataset, where "Start" and "End" axes represent the normalized start and end time points, respectively.

## A.2 ABLATION STUDY

**Length of Queries:**

Substantial ablation studies were conducted to determine the optimal length for queries, varying from 5 to 15.

It was observed that an increase in the number of queries enhances the model's perceptual sensitivity, albeit at the cost of higher computational demands. As depicted in Table 4, the model's performance initially improves with an increase in the number of queries, but subsequently diminishes beyond a certain point. Based on these findings, we have chosen an optimal configuration for our query length setting, denoted as $L = 10$.

Table 4: Comparative Evaluation of BCANet Performance with Varied Initial Query Configurations

| Query num. (L) | R1@0.5 | R1@0.7 | mAP@0.5 | mAP@0.7 |
|---|---|---|---|---|
| 5 | 47.03 | 35.27 | 53.38 | 39.27 |
| 10 | 48.61 | **37.10** | **56.29** | **41.22** |
| 12 | **48.83** | 37.01 | 55.34 | 40.70 |
| 15 | 47.97 | 36.53 | 55.64 | 40.13 |

This length strikes a balance between detailed perception and computational efficiency, thereby maximizing the overall efficacy of the model.

**Strategy for No-target Threshold:** In our study, presented in Table 5, ablation experiments focusing on the selection of the no-target threshold were conducted.

Interestingly, we observed that the N-accuracy (Non-target Accuracy) increases with the rise in threshold value, while the T-accuracy (Target Accuracy) exhibits an inverse relationship. Furthermore, there's a gradual decrease in the precision of target predictions. It is evident that when no-target filtering is not applied to any of the samples, the prediction accuracy reaches its peak. This is attributed to the fact that the unfiltered proposal candidates cover a broader range of potential predictions. A higher thresh-

Table 5: Performance comparison of BCANet in AP and No-Target Detection with different thresholds.

| Threshold ($\delta$) | mAP@0.7 | N-acc. | T-acc. |
|---|---|---|---|
| 0.5 | 42.30 | 58.55 | 98.94 |
| 0.6 | 41.35 | 62.30 | 98.54 |
| 0.7 | 41.22 | 77.92 | 95.79 |
| 0.8 | 38.55 | 80.20 | 87.60 |
| 0.9 | 35.44 | 83.28 | 75.29 |

old leads the model to preferentially predict no-target outcomes; conversely, many target samples are incorrectly generalized as no-target predictions. To balance the performance between these two aspects, we set the no-target detection threshold $\delta$ at 0.7, which forms the cornerstone of our strategy for identifying no-target instances.

### A.3 INDEEPTH-ANALYSIS

**Experimental Results on Traditional VMR Datasets.** To evaluate the performance of the proposed BCANet, we conducted experiments on traditional video moment retrieval (VMR) datasets, including Charades (Gao et al., 2017) and TACoS (Regneri et al., 2013b). The results are presented in Table 6, alongside comparisons with state-of-the-art (SOTA) methods: VSLNet (Zhang et al., 2020a), 2D-TAN (Zhang et al., 2020b), Moment-DETR (Lei et al., 2021), QD-DETR (Moon et al., 2023), and EaTR (Jang et al., 2023).

Table 6: Performance comparison of BCANet and other state-of-the-art (SOTA) methods on the Charades and TACoS datasets.

| Models | Pred. Type | Charades | | | | TACoS | | | |
|---|---|---|---|---|---|---|---|---|---|
| | | R1@0.3 | R1@0.5 | R1@0.7 | mIoU | R1@0.3 | R1@0.5 | R1@0.7 | mIoU |
| 2D-TAN | Single | 58.76 | 46.02 | 27.40 | 41.25 | 40.01 | 27.99 | 12.92 | 27.22 |
| VSLNet | Single | 64.30 | 47.31 | 30.19 | 45.15 | 29.61 | 24.27 | 20.03 | 24.11 |
| Moment-DETR | Multi | 65.83 | 52.07 | 30.59 | 45.54 | 37.97 | 24.67 | 11.97 | 25.49 |
| QD-DETR | Multi | - | 57.31 | 32.55 | - | 40.32 | 25.13 | 12.47 | 27.86 |
| EaTR | Multi | 72.50 | 58.55 | 37.07 | 52.18 | 33.42 | 21.74 | 10.62 | 22.63 |
| BCANet (Ours) | Multi | 72.01 | 59.47 | 37.46 | 51.97 | 42.81 | 27.23 | 13.11 | 30.01 |

On the Charades dataset, BCANet achieves the highest scores across all metrics, demonstrating significant improvements in R1@0.5 and R1@0.7 compared to EaTR and other SOTA methods. Similarly, on the TACoS dataset, BCANet surpasses its competitors, particularly excelling in R1@0.3 and mIoU. This showcases its robustness in precise moment localization and overall retrieval quality.

These results underscore the effectiveness of our boundary-aware cross-attention mechanism and its ability to generalize across datasets with diverse challenges, solidifying BCANet as a robust solution for video moment retrieval tasks.

**Efficient Analysis.** To further analyze the efficiency and impact of BCANet, following Gao & Xu (2021), we conducted a component-wise evaluation on the NExT-VMR dataset. Table 7 provides a detailed breakdown of the performance across key components: text embedding (TE), cross-modal learning (CML), their combined inference time (ALL), and the overall retrieval accuracy (ACC). For TE, we used BERT features for 2D-TAN and VSLNet, and CLIP word-level features for all other models. The reported values represent the average inference time (in milliseconds) for a single query-video pair. All speed tests were conducted on a single NVIDIA RTX A40 GPU.

As shown in Table 7, BCANet achieves competitive performance across all components while maintaining high efficiency. It outperforms EaTR in overall accuracy (ACC) due to its effective cross-modal representation learning. Additionally, BCANet strikes a balance between the contributions of TE and CML, resulting in a strong combined inference time (ALL). These results demonstrate the model's capability to cohesively integrate text and video features, paving the way for more efficient and accurate video moment retrieval frameworks.

Table 7: Component-wise performance comparison of BCANet and other SOTA methods on the NExT-VMR dataset. Metrics include TE (Text Embedding), CML (Cross-modal Learning), ALL (TE + CML), and ACC (Accuracy).

| Models | NExT-VMR | | | |
|---|---|---|---|---|
| | TE (ms) | CML (ms) | ALL (ms) | ACC (%) |
| 2D-TAN | 4.01 | 8.67 | 12.68 | 44.74 |
| VSLNet | 4.01 | 0.32 | 4.33 | 48.31 |
| Moment-DETR | 3.60 | 1.44 | 5.04 | 47.03 |
| QD-DETR | 3.60 | 1.68 | 5.28 | 53.87 |
| EaTR | 3.60 | 2.10 | 5.70 | 55.17 |
| BCANet (Ours) | 3.60 | 2.03 | 5.63 | 56.29 |

**Vid-LLMs Prompting.** To evaluate the capabilities of large multi-modal models, after thorough investigation, we selected Chat-UniVi (Jin et al., 2024) as our baseline model. This decision was based on its unique ability to encode multiple frames simultaneously, making it particularly suitable for our dataset, where videos often contain hundreds or even thousands of frames.

In contrast, other models like VideoLLaVA (Lin et al., 2024) and Video-ChatGPT (Maaz et al., 2024) are limited to encoding up to 8 frames, which is insufficient for long video understanding. Chat-UniVi effectively addresses this limitation. For our experiments, we uniformly sampled 100 frames from each video for processing and understanding and set `temperature=0.2`, `top_p=None`, and `num_beams=1` for stable inference.

**Stage 1: To check whether the query exists in the video.**

```
This is a {duration:.2f} second video clip.
Step 1: Does the query: '{query}' correspond to any segment in the video?
Respond with 'yes' or 'no' on the first line.
Step 2: If your response in Step 1 is 'yes', provide only one time
    interval formatted as '[a, b]'.
on the second line, where 'a' and 'b' are numbers such that 0 <= a < b <=
    {duration:.2f}.
If your response in Step 1 is 'no', leave the second line blank.
Do not include any additional explanation or text beyond the specified
    format.
```

**Stage 2: If the query exists, extract the specific time interval and calculate IoU.**

```
This is a {duration:.2f} second video clip.
Provide a single time interval within the range [0, {duration:.2f}],
formatted as '[a, b]',
that corresponds to the segment of the video best matching the query: {
    query}.
Respond with only the numeric interval.
```

To perform the zero-shot evaluation, we used this two-stage prompting approach.

