# OpenReview forum: "Generalized Video Moment Retrieval"
_ICLR.cc/2025/Conference — ICLR 2025 Poster_

### Official Review · Reviewer_LXSe · 2024-10-30

**Soundness:** 3
**Presentation:** 3
**Contribution:** 3
**Rating:** 6
**Confidence:** 5

**Summary:**

This paper introduces the Generalized Video Moment Retrieval (GVMR) framework, which expands traditional Video Moment Retrieval (VMR) to accommodate both non-target and multi-target queries, moving beyond the usual single-target focus. To support this, the authors present the NExT-VMR dataset, derived from the YFCC100M collection, featuring diverse query scenarios for robust model evaluation. They also propose BCANet, a transformer-based model that incorporates the Boundary-aware Cross Attention (BCA) module to enhance boundary detection and improve understanding of video content in relation to queries. Experiments demonstrate the potential of the NExT-VMR dataset and BCANet to advance VMR capabilities.

**Strengths:**

- This paper introduces a new task called Generalized Video Moment Retrieval (GVMR), which is designed to adeptly handle the complexities of various query types, extending the traditional Video Moment Retrieval (VMR) paradigm to include both multiple-target and no-target queries.
- This paper presents a specialized dataset, NExT-VMR, which is meticulously constructed and analyzed from the YFCC100M dataset. Tailored specifically for GVMR, this dataset features a diverse range of query types, including one-to-multiple and no-target queries.
- This paper introduces a model called BCANet, aimed at improving performance on the challenging task of GVMR, where the model must accurately predict a variable number of temporal timestamps based on natural language descriptions.

**Weaknesses:**

- Why does the paper utilize the YFCC100M dataset as the primary data source instead of existing VMR datasets such as TACoS, Charades-STA, MAD, or ActivityNet-Captions?
- From the statistics in Table 1, it can be seen that the average number of segments corresponding to a query is less than 2, which contradicts the paper's motivation for creating multiple-target queries. Additionally, the proportion of non-target queries is very low. Based on the dataset statistics, it is unclear where the challenges of this new dataset compared to existing ones lie.
- From Table 2, it can be seen that BCANet shows a significant advantage only in the single-moment retrieval task, while its performance in multi-target and non-target scenarios is not as pronounced. This raises questions about the overall effectiveness of the method.

**Questions:**

- BCANet is a supervised learning framework. How does it address the issue of inconsistent target quantities for each query in a batch during the training process? Additionally, if a query corresponds to three moments, are all three segments' timestamp information used in the loss function calculation?
- For queries labeled as non-target, what information do they provide during training?
- Why does the paper utilize the YFCC100M dataset as the primary data source instead of existing VMR datasets such as TACoS, Charades-STA, MAD, or ActivityNet-Captions?
- From the statistics in Table 1, it can be seen that the average number of segments corresponding to a query is less than 2, which contradicts the paper's motivation for creating multiple-target queries. Additionally, the proportion of non-target queries is very low. Based on the dataset statistics, it is unclear where the challenges of this new dataset compared to existing ones lie.
- From Table 2, it can be seen that BCANet shows a significant advantage only in the single-moment retrieval task, while its performance in multi-target and non-target scenarios is not as pronounced. This raises questions about the overall effectiveness of the method.

---

> ### Author Response · Authors · 2024-11-23
>
> We thank the reviewer for the positive feedback on our benchmark. We reply to each concern below:
>
> **W1: Good question**. The primary challenge in the current VMR community is that existing datasets like ActivityNet-Caption, Charades, and TACoS are designed for short videos with simple actions. However, these datasets have several limitations:
>
> 1. **Scale and Complexity**: These datasets contain relatively few videos, and the events or actions within the videos are often singular and straightforward. This simplicity limits their ability to simulate the complexities and diversity of generalized video moment retrieval (GVMR) tasks, which require handling everyday, dynamic, and varied scenarios.
> 2. **Lack of Progress on Standard Benchmarks**: Despite years of research, these datasets have been extensively studied, and models like BAN-APR [1] (stops at 43% for IoU@0.7 metric for years) have achieved strong performance. However, there have been limited breakthroughs in performance on these benchmarks in recent years. This stagnation highlights the need for a larger, semantically rich dataset to drive further advancements in GVMR research.
>
> While the MAD dataset offers semantic richness, it primarily consists of curated movie scenes. This focus on narrative-driven and meticulously designed settings makes it more suitable for tasks like causal inference in storytelling, which, while relevant, is only a subset of the broader VMR research community's needs. Additionally, ethical concerns surrounding the use of movies further complicate its applicability.
>
> In contrast, the YFCC100M dataset offers significant advantages:
>
> - **Scale and Diversity**: It contains a massive number of videos, with varied lengths and rich semantic information, covering a wide range of motions, events, and everyday scenarios.
> - **Generalization**: Its diversity in content makes it an ideal choice for developing and evaluating models capable of handling generalized video moment retrieval in real-world scenarios.
>
> These characteristics make YFCC100M the preferred choice for our work, as it aligns closely with the need for a large-scale, semantically diverse dataset to advance GVMR research.
>
> **W2:** We appreciate the observation. While the average number of segments corresponding to a query is less than 2, this figure includes no-target cases, which lowers the overall average. For queries that do have multiple targets, the actual average number of segments is 3. As shown in Figures 4, 5, and 6 in the Appendix—particularly Figures 4(a) and 4(b)—our dataset features a relatively balanced proportion of multi-target queries and includes a meaningful number of no-target queries. These figures provide a more comprehensive perspective on the dataset’s structure and highlight the challenges it introduces compared to existing datasets.
>
> **W3:**  In Table 2, the metrics **mAP@K**, **N-acc**, and **T-acc** are specifically designed to evaluate performance in multi-target and non-target scenarios:
>
> •	**mAP@K** measures the model’s ability to handle multi-query grounding. A segment is considered grounded if its IoU with a ground truth segment exceeds a threshold. The average precision (AP) for a single query is calculated as the ratio of the number of grounded segments to the total number of ground truth segments, reflecting the model’s effectiveness in multi-target scenarios.
>
> •	**N-acc** evaluates whether a non-target segment is correctly classified as a non-target sample.
>
> •	**T-acc** measures whether a segment labeled with a correct query is accurately grounded as a target sample.
>
> Our model achieves **SOTA performance** across all these metrics compared to open-source solutions, demonstrating its robustness and effectiveness in both multi-target and non-target scenarios.

---

> > ### Author Response · Authors · 2024-11-23
> >
> > **Q1:** In one-to-many matching, the issue of varying target quantities per query is typically addressed using **dynamic matching mechanisms**. These mechanisms dynamically assign targets to queries based on similarity metrics, such as IoU or feature distance. Specifically, in BCANet, **Equation 8** addresses this issue by dynamically matching queries with their corresponding targets during training. This ensures that each query is flexibly and adaptively matched with the most relevant targets, even when the number of targets varies for each query in a batch.
> >
> > If a query corresponds to three moments (or segments), all these segments are used during training. Each segment backpropagates through **different decoder queries**, which have the best match for that segment during the training process, which is presented in **Equation 8**. The matching process is dynamic and may change across iterations as the model parameters are updated. However, through continuous optimization, the matching results eventually converge to a stable state, allowing the model to effectively learn the relationships between queries and their corresponding targets.
> >
> > **Q2:** During training the proxy loss and class loss ($\mathcal{L}\_{proxy}$  and $\mathcal{L}\_{moment}$) are designed to provide whether the decode query are corresponded a non-target language-video pair. However, the development of a robust and effective mechanism for non-target compression remains an open-ended challenge. We encourage further exploration and more sophisticated designs in this area as part of future work.
> >
> > **Q3,4,5: Please refer to W 1,2,3.**
> >
> > [1] Wang J, Ma L, Jiang W. Temporally grounding language queries in videos by contextual boundary-aware prediction[C]//Proceedings of the AAAI Conference on Artificial Intelligence. 2020, 34(07): 12168-12175.

---

### Official Review · Reviewer_4Qqv · 2024-11-01

**Soundness:** 2
**Presentation:** 2
**Contribution:** 3
**Rating:** 6
**Confidence:** 5

**Summary:**

This paper introduces the Generalized Video Moment Retrieval framework, which expands traditional VMR to handle complex, non-target, and multi-target queries. To support this, the authors present the NExT-VMR dataset and BCANet, a transformer-based model with a Boundary-aware Cross Attention module that improves boundary detection and query understanding.
The results show that GVMR, NExT-VMR, and BCANet significantly enhance VMR performance, setting a new benchmark for future research in multimedia information retrieval.

**Strengths:**

The problem the authors solve is interesting.

This paper provides a good moment retrieval dataset.

The paper is clearly written and the method can be understood.

**Weaknesses:**

- The NExT-VMR dataset is not large enough to be used for large-scale training. In addition, why not divide the test set on NExT-VMR for experiments?

- Figure 3 is confusing. There are many colored blocks around the Query-Text Cross Attention module. I don't understand what it wants to express? Lack of technical details.

**Questions:**

- Can't the previous model achieve multi-target query well after training on the standard dataset? Lack of relevant experiments to prove the author's motivation.

- Can the model in this paper use a single query to query multiple related clips in the video? This is a more worthy of attention.

- YFCC100M is an excellent large-scale video dataset, but why did the author only select 9957 videos from YFCC100M? How was it selected?
- Will the author open source the dataset and code?

---

> ### Author Response · Authors · 2024-11-23
>
> We appreciate the reviewer's positive feedback on our model, training pipeline, comparisons with baselines, and its versatility on related tasks. We address your concerns below:
>
> **W1:**
>
> **Firstly,** we compare our dataset with other mainstream datasets for the video moment retrieval (VMR) task. As shown in Table 1, our dataset contains 9,957 videos, which is on par with other datasets (ranging from 127 to 14,926 videos). Furthermore, it includes 153,191 queries, surpassing most VMR datasets (which typically contain fewer than 100,000 queries, except for MAD). Notably, none of the existing datasets support complex query types, such as multi-target or no-target queries simultaneously. Additionally, we conducted a query length comparison with other VMR datasets, as illustrated in Figure 5(b) in the Appendix. Our dataset features a much larger proportion of longer queries compared to TACoS_2DTAN, Charades-STA, and Activity-Net Caption. Specifically, most of our queries range from 10 to 40 words in length, whereas the majority of queries in other datasets are within 30 words.
>
> **Secondly,** we have provided train, validation, and test splits for the NExT-VMR dataset. Figure 4 in the Appendix illustrates the distribution of queries across these splits, with 107,589 queries in the training set, 11,808 in the validation set, and 33,794 in the test set. Additionally, we also provide query annotation distributions for the train, validation and test sets in Figure 6. These distributions share similar characteristics, showing moments widely spread across the upper triangle, despite many being concentrated in the upper-left region.
>
> **W2:** We apologize for the confusion regarding Figure 3. In the updated figure, the different colors represent distinct, non-overlapping segments with different ground truth labels that are to be grounded in the video. These colors visually differentiate segments corresponding to specific queries. To address the issue, we have revised and clarified the figure in the updated PDF, ensuring it provides a more accurate and detailed representation of our model.
>
> **Q1**: Yes, we have implemented all publicly available methods as baselines for multi-target prediction and adapted them to include non-target detection evaluation. As shown in Table 2, while these methods demonstrate a certain level of performance, our specially designed modules offer notable improvements, showcasing the effectiveness of our approach.
>
> To further evaluate performance on standard datasets, we implemented two one-to-one baseline models. The results indicate that while these models perform adequately on single-target queries, their performance on multi-target metrics is notably lower compared to specifically designed multi-target models like BCANet. This highlights the necessity of specialized designs for multi-target video moment retrieval tasks.
>
> | Models | mAP (%) |
> | --- | --- |
> | 2D-TAN | 44.74 |
> | VSLNet | 48.31 |
> | Moment-DETR | 47.03 |
> | QD-DETR | 53.87 |
> | EaTR | 55.17 |
> | **BCANet (Ours)** | **56.29** |
>
> **Q2:** Yes, our model is specifically designed to support querying multiple related clips in a video using a single query. It effectively handles one-to-one, one-to-many, and even one-to-none query scenarios. This functionality aligns with our core motivation to address diverse grounding tasks, making the model well-suited for retrieving and grounding multiple related clips for a single query.
>
> **Q3:**  Good catch! We specifically chose 9,957 videos from YFCC100M based on the following considerations:
>
> - **Appropriate Duration**: The videos in our dataset have an average duration of 35.54 seconds, striking a balance between capturing meaningful information and avoiding unnecessary computational complexity. This makes the dataset well-suited for video-related tasks.
> - **Diversity and Scale**: We carefully removed videos with repetitive scenes to enhance diversity and ensure that each video contributes unique content. This approach maintains the complexity of the dataset while increasing its variety. As shown in Table 1, the resulting dataset is also comparable in scale to other widely-used Video Moment Retrieval (VMR) datasets, making it robust enough for comprehensive experiments and fair comparisons with existing benchmarks.
> - **Future Expansion Plans**: While this version of the dataset meets current research needs, we plan to expand it further in the future to provide the community with an even larger and more diverse resource for advancing video understanding tasks.
>
> By carefully selecting videos based on these criteria, we ensured the dataset aligns well with the requirements of our experiments while maintaining high quality and relevance. We hope this clarifies your concerns.
>
> **Q4:** Yes, we will release our dataset and code once accepted.

---

> > ### Comment · Reviewer_4Qqv · 2024-11-25
> >
> > The author addressed my concerns well, so I'm raising the score to marginally above the acceptance threshold

---

> > > ### Author Response · Authors · 2024-11-25
> > >
> > > Thank you for your support! We're glad to hear your concerns have been resolved. We always appreciate your suggestions and feedback!

---

### Official Review · Reviewer_Zk91 · 2024-11-02

**Soundness:** 3
**Presentation:** 3
**Contribution:** 3
**Rating:** 6
**Confidence:** 5

**Summary:**

This paper introduces the Generalized Video Moment Retrieval (GVMR) framework, which extends traditional Video Moment Retrieval (VMR) to handle a wider range of query types. To support this expanded task, it presents the NExT-VMR dataset, derived from the YFCC100M collection, featuring diverse query scenarios to enable more robust model evaluation. Additionally, a novel transformer-based model named BCANet is proposed and achieves outperforms when compared with traditional VMR approaches.

**Strengths:**

1. A novel GVMR framework is first introduced that significantly expands the scope of traditional VMR, catering to a wider range of query types and enhancing the model's practical utility.

2. A meticulously curated dataset, NExT-VMR, is presented. The dataset is specifically designed for GVMR task, which includes a variety of query types and scenarios, thereby enabling more robust and versatile model training and evaluation.

**Weaknesses:**

1. As mentioned in the paper, "GVMR aspires to deliver a robust and flexible solution for video moment retrieval, significantly enhancing the model's interpretative prowess, precision, and retrieval efficiency." Would the enhancement of the model on interpretative prowess, precision, and retrieval efficiency be proved and presented in experiments?

2. More statistics of the proposed dataset are needed to be listed in Table 1, such as the entire/average lengths of videos and queries and the number of one-to-multi and no-target samples in the dataset.

3. Is it possible to demonstrate the evaluation of this method on traditional VMR datasets (e.g., Charades, ActivityNet, and TACoS) for one-to-one (different from the mentioned one-to-multi or no-target) VMR tasks?

4. The BCANet pipeline presented in Figure 1 is confusing and difficult to interpret. The respective variable symbols should be labeled in Figure 1. Moreover, the processes for the BCL and QCA modules, along with query-text cross-attention, are also not clearly depicted in the figure. Moreover, what's the meaning of the "region" and "region features" in this paper? Dose the "region" mean a video clip?

5. The BCANet proposed in this paper does not seem to include specific designs for handling one-to-multi cases. Compared to traditional VMR methods, what advantages does BCANet offer in dealing with such data?

6. Since the retrieval speed is also an important metric in VMR, for example, [a] analyze the speed of various VMR approaches. I recommend the authors conducting speed analysis in this paper. In addition, is GVMR slower than traditional VMR?
[a] Fast video moment retrieval, ICCV 2021.

**Questions:**

Please refer to the weakness part.

---

> ### Author Response · Authors · 2024-11-23
>
> We thank the reviewer for their positive feedback on our task, model, and results. We address the weaknesses below.
>
> **W1:** Yes, we have addressed your concerns in the subsequent sections of the paper. The enhancements in the model's efficiency, interpretative prowess, and precision are demonstrated and analyzed in our experiments. Specifically:
>
> - **Interpretative prowess and precision** refer to the model's robustness in understanding the query, which is primarily reflected in metrics such as IoU and mAP. These results are detailed in the main results table, Table 2.
> - **Efficiency** is further analyzed Table 7. on component-wise evaluations, where we break down the inference times of key modules to highlight the model's performance in retrieval speed.
>
> For comprehensive details, please refer to Table 2 and the accompanying discussions.
>
> **W2:** You may find more detailed statistics in Appendix: Figure 4 about the distribution of (a) query numbers relevant to N clips and (b) Query Type; Figure 5 about (a) Statistics Distribution for Query Number and Each Corresponding Segments of Ground Truth Timestamps in Log Scale. (b) Distribution for Query Length with comparison with other datasets; Figure 6 about Moment annotation distributions of NExT-VMR dataset, where “Start” and “End” axes represent the normalized start and end time points, respectively. Additionaly, the average video duration length is 35.54 seconds.
>
> **W3**:  Yes, we evaluated BCANet on traditional VMR datasets, Charades and TACoS, comparing it against all publicly available multi-prediction models. BCANet outperformed these models in one-to-one VMR tasks, as shown in Table 1, highlighting its strong adaptability and superior performance in this context.
>
> ---
>
> ### Table 1: Performance Comparison on Charades and TACoS Datasets
>
> | Models | Pred. Type | Charades R1@0.3 | Charades R1@0.5 | Charades R1@0.7 | Charades mIoU | TACoS R1@0.3 | TACoS R1@0.5 | TACoS R1@0.7 | TACoS mIoU |
> | --- | --- | --- | --- | --- | --- | --- | --- | --- | --- |
> | 2D-TAN | Single | 58.76 | 46.02 | 27.40 | 41.25 | 40.01 | 27.99 | 12.92 | 27.22 |
> | VSLNet | Single | 64.30 | 47.31 | 30.19 | 45.15 | 29.61 | 24.27 | 20.03 | 24.11 |
> | Moment-DETR | Multi | 65.83 | 52.07 | 30.59 | **45.54** | 37.97 | 24.67 | 11.97 | 25.49 |
> | QD-DETR | Multi | - | 57.31 | 32.55 | - | 40.32 | 25.13 | 12.47 | 27.86 |
> | EaTR | Multi | **72.50** | 58.55 | 37.07 | **52.18** | 33.42 | 21.74 | 10.62 | 22.63 |
> | BCANet (Ours) | Multi | 72.01 | **59.47** | **37.46** | 51.97 | **42.81** | **27.23** | **13.11** | **30.01** |
>
> ---

---

> > ### Author Response · Authors · 2024-11-23
> >
> > **W4:** Thank you for pointing out these issues. In the updated version of the paper, we have updated the main diagram of our model to improve clarity. The revised figure now includes labelled variable symbols and depicts the processes for the BCL and QCA modules, as well as the query-text cross-attention mechanism.
> >
> > Regarding the terms "region" and "region features" in our paper, a "region feature" refers to the feature obtained after applying average pooling in the BCL module. This feature represents the active foreground video features defined by the ground truth boundary labels. It is distinct from the term "video clip," as it specifically focuses on the feature representation within the labeled region rather than the raw video segment.
> >
> > **W5**: Our proposed BCANet tackles challenges in one-to-many cases by introducing a novel contrastive learning approach at the fusion stage of language and video modalities. By ensuring that decoder queries—fused representations of language and video inputs—exhibit higher affinity (i.e., smaller distances in the contrastive feature space) to the time spans of interest (SoI) through features averaged within boundaries, our method integrates boundary information effectively during training. This design enhances sensitivity across diverse SoIs, allowing the model to focus on relevant temporal segments, while also improving robustness by minimizing false positives in no-target scenarios. Unlike traditional methods that emphasize boundary prediction at the feature level, our approach incorporates boundary awareness more effectively at the fusion stage, leading to superior VMR performance in complex one-to-many scenarios. The ablation study (Appendix) further demonstrates the clear advantages of our method over traditional approaches.
> >
> > **W6:** Yes, retrieval speed is indeed a critical metric in Video Moment Retrieval (VMR), as highlighted in [a] (*Fast Video Moment Retrieval, ICCV 2021*). To address this, we have conducted a component-wise performance analysis of BCANet and other state-of-the-art (SOTA) methods on the NExT-VMR dataset.
> >
> > The results, presented in Table 2 below, evaluate key components, including Text Embedding (TE) (for 2D-TAN and VSLNet, we extract BERT [1] features, while for the rest, we use CLIP word-level features), Cross-Modal Learning (CML), the combined retrieval process (ALL), and mean Average Precision (mAP ). The times are reported in milliseconds (ms), representing the average inference time for a single query-video pair for each model. BCANet achieves a total retrieval time (ALL) of 5.63 ms and the highest accuracy on multi-target (mAP) of 56.29%, showcasing its ability to balance efficiency and effectiveness.
> >
> > Regarding GVMR, while global video moment retrieval may introduce additional computational complexity compared to traditional methods, our analysis demonstrates that BCANet's optimized architecture effectively addresses this challenge. It delivers competitive retrieval speeds while maintaining superior accuracy.
> >
> > All speed tests were conducted on a single NVIDIA RTX A40 GPU.
> >
> > | Models | TE (ms) | CML (ms) | ALL (ms) | mAP (%) |
> > | --- | --- | --- | --- | --- |
> > | 2D-TAN | 4.01 | 8.67 | 12.68 | 44.74 |
> > | VSLNet | 4.01 | 0.32 | 4.33 | 48.31 |
> > | Moment-DETR | 3.60 | 1.44 | 5.04 | 47.03 |
> > | QD-DETR | 3.60 | 1.68 | 5.28 | 53.87 |
> > | EaTR | 3.60 | 2.10 | 5.70 | 55.17 |
> > | BCANet (Ours) | 3.60 | 2.03 | 5.63 | **56.29** |

---

> > > ### Comment · Reviewer_Zk91 · 2024-11-26
> > >
> > > The author's response has addressed my concerns, therefore I am maintaining my original positive score.

---

> > > > ### Author Response · Authors · 2024-11-26
> > > >
> > > > Thank you for your positive feedback. We are glad that our response has addressed your concerns.

---

### Official Review · Reviewer_feEk · 2024-11-07

**Soundness:** 3
**Presentation:** 3
**Contribution:** 3
**Rating:** 6
**Confidence:** 4

**Summary:**

This paper first introduce a novel GVMR framework that significantly expands the scope of traditional VMR, catering to a wider range of query types and enhancing the model’s practical nutility and  present a meticulously curated dataset. To deal with GVMR, they propose the BCANet model. Totally speaking, the contributions about extend task and new dataset is valuable.

**Strengths:**

1. This paper introduces a extend task about TVG which covers more situations about video localiztion.
2. They build a new dataset for their introduced dataset which helps the deeper researches in future.
3. They propose a model based on LLM generate queries in fine-grain semantic, to benefit the video content understanding to the next stage

**Weaknesses:**

1. The boundary awareness is a common question in this field and many works like[1] focus on this problem. The author didn't analyse the differences between them to show their novelty.


*[1]J. Wang, L. Ma, and W. Jiang, “Temporally grounding language queries in videos by contextual boundary-aware prediction,” in AAAI, 2020.

**Questions:**

1. The model relies on the LLaVa to generate scene captions to divide the backgrounds and foregrounds. Will it brings noisy due to the diversity of generated samples?
2. Did you compare your model with the finetuinng model based on multimodal LLM ?
3. With using the LLM to generate captions , what about the time cost and memory usage?
4. Will the dataset be open source? If so ,it will be an important contribution.

---

> ### Author Response · Authors · 2024-11-23
>
> We thank the reviewer for the positive comments on our task and our extensive experiments. We address the concern below:
>
> **W1**: The method you mentioned from the paper *“Temporally Grounding Language Queries in Videos by Contextual Boundary-Aware Prediction”* [1] utilizes clip-level features to predict boundary features through binary classification, as defined by the loss function:
>
> $$
> \mathcal{L}_{b}(t, X, z) {t}) = w\_{pos} \cdot z\_{t} \log{b\_t} + w\_{neg} \cdot (1 - z\_{t}) \log(1 -  b\_{t})
> $$
>
>
> This approach enhances the model’s prediction by conducting boundary-aware binary metric learning directly on the features before grounding MLPs.
>
> In contrast, our method introduces a novel approach by applying contrastive learning at the video-language fusion stage. We leverage contrastive learning to ensure that our decoder queries, which are fused representations of language and video inputs, have a higher affinity (measured by smaller distances in the contrastive feature space) to the time spans of interest (**SoI**). This is achieved by incorporating boundary information—obtained through the average pooling of features within the boundaries—into each mini-batch during training.
>
> Our design offers several key advantages:
>
> 1.	**Uniform Awareness Across Different SoIs**: The decoder queries become uniformly attentive to various SoIs, enhancing the model’s ability to accurately focus on relevant temporal segments.
>
> 2.	**Robustness to No-Target Queries**: In situations where there is no SoI (queries without a target ground truth), our decoder queries appropriately refrain from responding to any features, thereby improving the model’s robustness and reducing false positives.
>
> By integrating boundary awareness through contrastive learning at the fusion stage, our method differs from previous works that focus on boundary prediction at the feature level. This fusion-stage approach not only improves model performance but also effectively handles scenarios that traditional boundary prediction methods may not address as efficiently.
>
> Based on your feedback, we have added a technical comparison with the method [1] you mentioned in the revised version of our paper in sec. 4.2.
>
> [1] J. Wang, L. Ma, and W. Jiang, “Temporally grounding language queries in videos by contextual boundary-aware prediction,” *Proceedings of the AAAI Conference on Artificial Intelligence*, vol. 34, no. 7, pp. 12168–12175, 2020.

---

> > ### Author Response · Authors · 2024-11-23
> >
> > **Q1**: Yes, we did use LLaVa with a relatively high temperature=1.8 in **the first stage** to generate diverse scene captions for foreground identification. However, to ensure that the final query has minimal noise, we applied a lower temperature=0.2 in **the second stage** when using GPT to combine foreground and background, ensuring the generation stability. Additionally, in **the third stage**, we employed human verification to further validate the quality of the final generated queries.
> >
> > **Q2**: Yes, we compared our model’s performance with state-of-the-art large multimodal models for long video understanding, focusing specifically on zero-shot performance. After thorough investigation, we selected Chat-UniVi [1], a CVPR 2024 Highlight paper, as our baseline model. This decision was based on its unique ability to encode multiple frames simultaneously, making it particularly suitable for our dataset, where our videos often contain hundreds or even thousands of frames.
> >
> > In contrast, other models like VideoLLaVA (EMNLP 2024) [2] and Video-ChatGPT [3] are limited to encoding up to 8 frames or so, which are insufficient for long video understanding. Chat-UniVi [1] addresses this limitation effectively. For our experiments, we uniformly sampled 100 frames from each video for processing and understanding, and set temperature=0.2, top_p=None, num_beams=1 for stable inference.
> >
> > To perform the zero-shot evaluation, we used a two-stage prompting approach:
> >
> > ```markdown
> > # Stage 1: To check whether query exists in the video.
> > This is a {duration:.2f} second video clip.
> > Step 1: Does the query: '{query}' correspond to any segment in the video?
> > Respond with 'yes' or 'no' on the first line.
> > Step 2: If your response in Step 1 is 'yes', provide only one time interval formatted as '[a, b]'
> > on the second line, where 'a' and 'b' are numbers such that 0 ≤ a < b ≤ {duration:.2f}.
> > If your response in Step 1 is 'no', leave the second line blank.
> > Do not include any additional explanation or text beyond the specified format.
> > ```
> >
> > ```markdown
> > # Stage 2: If exists, we extract the specific time interval and calculate iou then.
> > This is a {duration:.2f} second video clip.
> > Provide a single time interval within the range [0, {duration:.2f}],
> > formatted as '[a, b]',
> > that corresponds to the segment of the video best matching the query: {query}.
> > Respond with only the numeric interval.
> > ```
> >
> > Using this approach, we effectively utilized Chat-UniVi to understand long videos and perform video moment retrieval tasks. The results of the zero-shot evaluation are provided below.
> >
> > | Model | Pred. Type | R1@μ = 0.3 | R1@μ = 0.5 | R1@μ = 0.7 | mIoU | mAP@μ = 0.25 | mAP@μ = 0.5 | mAP@μ = 0.75 | Avg. | N-acc. | T-acc. |
> > | --- | --- | --- | --- | --- | --- | --- | --- | --- | --- | --- | --- |
> > | Chat-UniVi*  | Prompt | 57.16 | 35.49 | 17.81 | 38.64 | 62.88 | 35.49 | 14.14 | 37.50 | - | - |
> > | Chat-UniVi  | Prompt | 46.15 | 27.52 | 13.81 | 29.62 | 42.13 | 23.71 | 9.95 | 25.26 | 71.55 | 68.48 |
> >
> > **Q3**: When using LVLMs and LLMs for caption generation, the first stage employs LLaVA-7B Int8, running on a single 4090 GPU with a max_length of 256 in bf16 precision. The inference time for a single prompt run is approximately 2~5 second, requiring 10–12 GB of memory. For the second stage, we directly used the GPT-3.5 API. As per OpenAI’s pricing, the cost is $0.50 per 1M input tokens and $1.50 per 1M output tokens. The overall expense for generating captions stayed within $100.
> >
> > **Q4**: **Yes, we will release our dataset and code upon acceptance.**
> >
> > [1] Peng Jin, Ryuichi Takanobu, Wancai Zhang, Xiaochun Cao, and Li Yuan. Chat-univi: Unified visual
> > representation empowers large language models with image and video understanding, 2024. URL
> > https://arxiv.org/abs/2311.08046.
> >
> > [2]  Bin Lin, Bin Zhu, Yang Ye, Munan Ning, Peng Jin, and Li Yuan. Video-llava: Learning united visual
> > representation by alignment before projection. arXiv preprint arXiv:2311.10122, 2023.
> >
> > [3]  Muhammad Maaz, Hanoona Rasheed, Salman Khan, and Fahad Shahbaz Khan. Video-chatgpt:
> > Towards detailed video understanding via large vision and language models. In Proceedings of the
> > 62nd Annual Meeting of the Association for Computational Linguistics (ACL 2024), 2024.

---

### Public Comment · ~Luo_Yinhang1 · 2024-11-19
**Ethical Concerns Regarding License Compliance and Data Usage in Derived Datasets from YFCC100M**

I have concerns regarding the ethical aspects of this paper's use of the YFCC100M dataset. The following points need clarification:

1. License Compliance:
   - The YFCC100M dataset comprises various media files with differing licenses, yet the paper does not specify how the data was selected from the original dataset. There is no clear explanation of whether the selected data complies with the original licenses, particularly regarding permissions for modification and display.
   - For example, in the paper, some images are displayed (e.g., Figure 1, 2). It is unclear whether these images are permitted to be displayed under their respective licenses.

2. Webscope License Agreement:
   - The YFCC100M dataset is distributed under the Webscope License Agreement, which explicitly prohibits the redistribution of modified or derivative datasets. Relevant sections include:
     - Section 2.6: "You will not display, reproduce, transmit, distribute, sell, publish, provide to third parties, incorporate into third-party materials, or use the Data except as specifically permitted in this TOU."
     - Section 2.5: "You will not perform any analysis, reverse engineering or processing of the Data or any correlation with other data sources that could be used to determine or infer personally identifiable information of any Verizon Media user(s or that violates any applicable law in any applicable jurisdiction."
   - The paper mentions modifications (e.g., creating the NExT-VMR dataset) and displays content from the dataset, but it does not explain whether such actions align with the Webscope License Agreement or whether explicit authorization was obtained.

3. Community Contribution:
   - If the NExT-VMR dataset derived from YFCC100M cannot be redistributed due to license restrictions, the contribution to the community would be limited. This issue warrants further discussion and transparency in the paper.

The authors should address the following:
- Clearly state how the data was selected and whether the selection complies with the original licenses of YFCC100M.
- Confirm whether the modifications, analysis, and display of data align with the Webscope License Agreement or if explicit permission was obtained.
- Discuss the implications of potential license restrictions on the accessibility of the derived dataset and its contribution to the community.

Without addressing these points, there are significant ethical concerns regarding the use and redistribution of the data in this work.

---

### Author Response · Authors · 2024-11-23
**General Response to Dataset Source and Licenses**

# Data Source and License Compliance

The YFCC100M dataset comprises metadata for approximately 99.2 million photos and 0.8 million videos from Flickr, all shared under various Creative Commons licenses [1][2]. While access to the dataset through Yahoo Webscope is governed by the Webscope License Agreement, the original media files are hosted on Flickr under their respective Creative Commons licenses. This is clearly stated on the YFCC100M dataset homepage (<https://www.multimediacommons.org/>), and we directly crawled the data from Flickr. The origin description is as follows:

> "The YFCC100M is the largest publicly and freely usable multimedia collection, containing the metadata of around 99.2 million photos and 0.8 million videos from Flickr, all of which were shared under one of the various Creative Commons licenses."

Furthermore, in the YFCC100M dataset papers (both versions [1][2]), there is no relevant description about dataset restrictions.

## Use of Creative Commons-Licensed Media

Our research utilized videos directly obtained from Flickr, strictly adhering to the terms of their respective Creative Commons licenses. These licenses permit the use and display of the media, provided that the conditions specified by each license are met, such as attribution and non-commercial use. The sections of the Webscope License Agreement, specifically Section 2.6 ("You will not display, reproduce…") and Section 2.5 ("You will not perform any analysis, reverse engineering or processing"), refer to restrictions related to Yahoo Webscope’s own proprietary data, not the data source we obtained from Flickr.

## Compliance with Licensing Terms

We did not use any additional annotations or metadata from the YFCC100M dataset that are restricted under the Webscope License Agreement. Our work involved only the media content available under Creative Commons licenses, ensuring full compliance with the permissions granted by the original content creators.

## Related Research Contributions

The YFCC100M dataset has served as the foundation for several impactful research contributions, further demonstrating its utility and importance in advancing the field. Notable examples include:

1. **"FairFace: Face Attribute Dataset for Balanced Race, Gender, and Age for Bias Measurement and Mitigation," published in WACV 2021**, which created a face attribute dataset aimed at bias measurement and mitigation [3].
2. **"Towards Open-ended Visual Quality Comparison," published in ECCV 2024**, exploring visual quality assessment tasks [4].
3. **"Panoptic Video Scene Graph Generation," published in CVPR 2023**, focusing on generating structured representations for video understanding [5].

These works have similarly derived additional datasets and insights using YFCC100M as their starting point, aligning with the same ethical and licensing principles. It is precisely due to the open availability of YFCC100M that the dataset has been so successful and has garnered over 2,000 citations to date.

## Display of Images in the Paper

The images displayed in our paper (e.g., Figures 1 and 2) are used in accordance with the terms of their respective Creative Commons licenses. Similar to [3][4][5][6], we have ensured that we have the right to display these images for academic and research purposes, following all necessary attribution requirements.

## Contribution to the Community

Acknowledging the licensing restrictions, we will release only the video features extracted from the Creative Commons-licensed media and the annotations we have created ourselves. This approach allows us to contribute valuable resources to the community while respecting all legal and ethical guidelines.

## Ethical Considerations

Dear Yinhang,

As we have explained above, we have thoroughly evaluated the ethical aspects of our work and have consulted with the appropriate authorities to ensure compliance with all relevant policies and licenses. We are committed to upholding ethical standards in our research and fostering a responsible research environment.

Due to anonymous policy restrictions, we can only provide the information above to address your concerns. Additionally, we have submitted confidential comments to the Program Chairs and Area Chairs, and we believe there are no ethical concerns in our work.

Thank you again for your thoughtful feedback. We are open to further discussions to clarify any additional concerns and are committed to transparency in our research practices.

---

## References

[1] YFCC100M: The New Data in Multimedia Research. arXiv:1503.01817
[2] YFCC100M: The New Data in Multimedia Research. Communications of the ACM, Volume 59, Issue 2. 2016
[3] FairFace: Face Attribute Dataset for Balanced Race, Gender, and Age for Bias Measurement and Mitigation. WACV 2021.
[4] Towards Open-ended Visual Quality Comparison. ECCV 2024.
[5] Panoptic Video Scene Graph Generation. CVPR 2023.

---

> ### Public Comment · ~Luo_Yinhang1 · 2024-11-25
>
> Thank you for your detailed response. I have two key concerns that I would like to highlight:
>
> 1. Data Source Transparency: In your response, you mention that the media files were directly crawled from Flickr under their respective Creative Commons licenses. However, the paper explicitly states that the dataset is derived from YFCC100M, without clarifying that the media files were obtained from Flickr. This omission creates ambiguity for readers regarding the actual source of the data.
>
> 2. Compliance with the Webscope License Agreement: The paper explicitly clarify the annotations (Query statements) used in your work rely on YFCC100M. If your annotations are derived from YFCC100M metadata, this would invoke the Webscope License Agreement, which prohibits redistribution of modified datasets or annotations without explicit permission. If this is the case, please provide evidence of permission or authorization from Yahoo for such use.
>
> Addressing these points will help ensure transparency and compliance with licensing terms. Thank you for your attention, and I look forward to your response.

---

> ### Author Response · Authors · 2024-11-25
>
> **Data Source Transparency**
>
> > "Data Source Transparency: In your response, you mention that the media files were directly crawled from Flickr under their respective Creative Commons licenses. However, the paper explicitly states that the dataset is derived from YFCC100M, without clarifying that the media files were obtained from Flickr. This omission creates ambiguity for readers regarding the actual source of the data."
>
> We have already clarified the data source in our previous responses. To reiterate, all raw video files used in our research were directly crawled from Flickr under their respective Creative Commons licenses. This has now been explicitly stated in both the revised PDF version of the manuscript and in our earlier explanations.
>
> **Compliance with the Webscope License Agreement**
>
> > "The paper explicitly states that the annotations (query statements) used in your work rely on YFCC100M. If your annotations are derived from YFCC100M metadata, this would invoke the Webscope License Agreement, which prohibits redistribution of modified datasets or annotations without explicit permission. If this is the case, please provide evidence of permission or authorization from Yahoo for such use."
>
> We would like to clarify that no annotations from the YFCC100M dataset were used in our work. The annotations in our study were created through a rigorous process **with great efforts**, which has been explained in detail in Figure 2 and the Appendix of our paper. Thus, there was no reliance on YFCC100M metadata for the creation of our query statements.
>
> To ensure absolute transparency and address the concerns raised, we would like to emphasize the following:
>
> 1. All raw data (video content) used in our research was independently crawled from Flickr under their respective Creative Commons licenses.
> 2. To further address the ambiguity as you mentioned, we could have removed all mentions of YFCC100M in our paper. However, we believe this is not the ideal solution, as YFCC100M is a foundational and highly respected dataset that has significantly contributed to multimedia research. Removing references to it would undermine its value and importance to the community.
>
> Dear Yinhang,
>
> We acknowledge YFCC100M as a well-designed and impactful dataset, widely referenced and built upon by other works such as [1], [2], and [3]. We believe it is appropriate and respectful to credit YFCC100M for its contributions to the field.
>
> If you are one of the original authors of YFCC100M, we would welcome the opportunity to engage in a direct and constructive discussion to address any concerns and ensure full compliance with the authors' intentions. If, however, you are not affiliated with the authorship of YFCC100M, we kindly request that these concerns be evaluated with consideration of the detailed clarifications provided above.
>
> **References**
>
> [1] FairFace: Face Attribute Dataset for Balanced Race, Gender, and Age for Bias Measurement and Mitigation. WACV 2021.
>
> [2] Towards Open-ended Visual Quality Comparison. ECCV 2024.
>
> [3] Panoptic Video Scene Graph Generation. CVPR 2023.

---

> > ### Public Comment · ~Luo_Yinhang1 · 2024-11-26
> >
> > Dear Authors,
> >
> > Thank you for your updated response and for clarifying the data source in the revised paper. I appreciate the effort to ensure transparency regarding the media files' origin.
> >
> > However, regarding the second point, I believe there may still be a misunderstanding. Specifically, in Section 3.3 of your paper, under **"DATA PREPARATION"**, you state:
> > > "We manually extracted (subject, predicate, object) relation tuples from the original YFCC100M video collections and converted them into grammatically correct sentences to serve as query statements."
> >
> > This description appears to indicate that YFCC100M metadata was indeed used to derive the query statements. Could you clarify how this aligns with your statement that "no annotations from the YFCC100M dataset were used"? If the relation tuples were extracted from YFCC100M metadata, this would invoke the Webscope License Agreement, which governs the use of YFCC100M metadata and annotations.
> >
> > I would appreciate it if you could provide further clarification on this point, as the current explanation seems to leave room for ambiguity that could be interpreted as a misalignment with licensing terms.
> >
> > Thank you for your attention to this matter.

---

> ### Author Response · Authors · 2024-11-26
>
> Dear Luo Yinhang,
>
> We’d like to make following explanation:
>
> The relation tuples and corresponding ground-truth timestamps were not derived from the YFCC100M metadata. In fact, such information does not exist in the metadata. In the previous statement, "extract" refers to our manual annotation process, where we watched the videos directly. This process involved human observation to identify basic action units within the videos and annotate the query bases and their corresponding time segments. To address this, we have updated the wording in the revised PDF to more precisely describe our contributions and eliminate ambiguity.

---

### Meta-Review · Area_Chair_RLwM · 2024-12-19

**Metareview:**

This paper proposed a new Generalized Video Moment Retrieval framework, which expands traditional VMR to handle complex, non-target, and multi-target queries. To facilitate research on this task,  a new dataset named NExT-VMR dataset, is derived from the YFCC100M collection, featuring diverse query scenarios to enable more robust model evaluation. The advantage of the method is that it defines a new task and establishes a relevant dataset, proposing a effective baseline model. The weakness is that some references, which were already published in some conference, were cited in arxiv format. Since all reviewers leaned to accept this paper, I recommend accept to it if there is not ethic problem.

**Additional Comments On Reviewer Discussion:**

The concerns raised by reviewers include insufficient details about dataset construction, lack of complexity analysis and some statistic information about dataset, lack of comparison with some related works etc. The authors’ rebuttals addressed these concerns well. Two reviewers raised their ratings, one reviewer maintained his positive ratings, and one reviewer didn’t provide final rating.

---

### Decision · Program_Chairs · 2025-01-22

Accept (Poster)